# The Interplay between Peripherin 2 Complex Formation and Degenerative Retinal Diseases

**DOI:** 10.3390/cells9030784

**Published:** 2020-03-24

**Authors:** Lars Tebbe, Mashal Kakakhel, Mustafa S. Makia, Muayyad R. Al-Ubaidi, Muna I. Naash

**Affiliations:** Department of Biomedical Engineering, University of Houston, Houston, TX 77204, USA; ltebbe@Central.UH.EDU (L.T.); mkakakhe@central.uh.edu (M.K.); msmakia@Central.UH.EDU (M.S.M.)

**Keywords:** peripherin 2, retinal degeneration, retina, tetraspanin, photoreceptor

## Abstract

Peripherin 2 (Prph2) is a photoreceptor-specific tetraspanin protein present in the outer segment (OS) rims of rod and cone photoreceptors. It shares many common features with other tetraspanins, including a large intradiscal loop which contains several cysteines. This loop enables Prph2 to associate with itself to form homo-oligomers or with its homologue, rod outer segment membrane protein 1 (Rom1) to form hetero-tetramers and hetero-octamers. Mutations in *PRPH2* cause a multitude of retinal diseases including autosomal dominant retinitis pigmentosa (RP) or cone dominant macular dystrophies. The importance of Prph2 for photoreceptor development, maintenance and function is underscored by the fact that its absence results in a failure to initialize OS formation in rods and formation of severely disorganized OS membranous structures in cones. Although the exact role of Rom1 has not been well studied, it has been concluded that it is not necessary for disc morphogenesis but is required for fine tuning OS disc size and structure. Pathogenic mutations in *PRPH2* often result in complex and multifactorial phenotypes, involving not just photoreceptors, as has historically been reasoned, but also secondary effects on the retinal pigment epithelium (RPE) and retinal/choroidal vasculature. The ability of Prph2 to form complexes was identified as a key requirement for the development and maintenance of OS structure and function. Studies using mouse models of pathogenic *Prph2* mutations established a connection between changes in complex formation and disease phenotypes. Although progress has been made in the development of therapeutic approaches for retinal diseases in general, the highly complex interplay of functions mediated by Prph2 and the precise regulation of these complexes made it difficult, thus far, to develop a suitable Prph2-specific therapy. Here we describe the latest results obtained in Prph2-associated research and how mouse models provided new insights into the pathogenesis of its related diseases. Furthermore, we give an overview on the current status of the development of therapeutic solutions.

## 1. Introduction

Tetraspanins represent a family of highly conserved membrane proteins involved in a variety of functions. These functions include membrane organization and compartmentalization, cell signaling, adhesion and migration [1,2,3,4]. Structurally, tetraspanins consist of a short cytoplasmic N-terminus, four transmembrane domains (TM), one small and one large extracellular loop (EC1 and EC2, respectively), as well as a short cytoplasmic C-terminus [3,5,6]. The EC2 loop can be further divided in a conserved and a variable region [7,8]. The highly conserved part of the EC2 loop was shown to mediate the dimerization of tetraspanin proteins, while the variable region is required for specific interactions of the tetraspanins with their variable interaction partners [7,8,9,10]. Tetraspanin interactions can be divided into three levels of interactions [1]. The primary level consists of direct associations between two tetraspanin proteins as well as between tetraspanin and a non-tetraspanin interacting partners, interactions that resist the treatment with strong detergents [1,4]. Binding of these primary complexes to each other represents the secondary level which is indirect and in some cases supported by palmitoylation [1,4,11]. In the third level, tetraspanins can indirectly associate with their interacting partners to form large insoluble complexes, which only resist mild detergents like CHAPS [1,4,12,13]. The ability of tetraspanins to interact with each other and with other partners allows them to form tetraspanin-enriched microdomains known as the tetraspanin web [4,14].

Peripherin 2 (Prph2, formally known as retinal degeneration slow, RDS) represents a photoreceptor-specific tetraspanin. It is a structural protein that is critical for the proper development of rod and cone outer segments (ROS and COS, respectively) and thus for vision [15,16]. The main function of Prph2, in the morphogenesis of ROS and COS, is to promote membrane curvature, flattening and fusion; processes that are required for the rim formation of the OS discs and lamella [15,17,18,19,20]. Like the other members of the tetraspanin family, Prph2 contains two loops, referred to as intradiscal loops (D1 and D2, respectively). The D2 loop is equivalent to the EC2 loop of other tetraspanins and mediates the bulk of interactions of Prph2 [21,22,23]. Prph2 interacts with another photoreceptor-specific tetraspanin, called rod outer segment membrane protein 1 (Rom1) [24]. Prph2 forms non-covalently associated homo- and hetero-tetramers with Rom1. These tetramers associate and form covalently linked intermediate complexes. In addition to that, Prph2 is able to form higher order complexes consisting of several Prph2 homo-tetramers. Interestingly, hetero-tetramers including Rom-1 are excluded from these higher order complexes [21,25,26]. Formation of these complexes is required for Prph2 to promote the development of rim domains essential for OS formation [18,27,28]. In line with its key role in OS formation, mutations in *PRPH2* are connected with a multitude of retinal diseases ranging from autosomal dominant retinitis pigmentosa (ADRP) to macular degeneration (MD) [29]. The associated phenotypes show a high degree of variability of onset and phenotype even between patients carrying the same mutation. In addition to the degeneration of the retina, some Prph2 related diseases were found to cause secondary effects on the retinal pigment epithelium (RPE) or the choroid [29,30,31].

This review describes the importance of Prph2 in the development of both ROS and COS. Recent data provided new insight into the steps necessary for OS morphogenesis and the role of Prph2 in orchestrating these events. Furthermore, we will summarize how the precise formation and regulation of Prph2/Rom1 complex affects the function of Prph2 in rods versus cones. Finally, we will discuss the animal models expressing pathogenic Prph2 mutations and provide a short summary about therapeutic approaches so far aimed to treat Prph2 related diseases.

## 2. The Role of Prph2 in Photoreceptor Outer Segment Morphogenesis

Since mice lacking Prph2 (*Prph2^−/−^*) fail to form ROS, it became clear that the major function of Prph2 is in the initialization and elaboration of ROS [32]. Recent data confirmed this function and identified the C-terminus of Prph2 as the motif responsible for the initialization of ROS formation [33,34]. The photoreceptor outer segment (OS) represents a highly specialized form of a modified primary cilium [35]. The initial steps in the development of the OS are similar to the development of other primary cilia, starting with the attachment of the ciliary vesicle to the basal body [36,37,38]. After this initial step, the elongation of the ciliary axoneme and the expansion of the plasma membrane happen in a synchronous fashion. A recent study showed that these initial steps in ciliogenesis take place in both wild type (WT) and *Prph2^−/−^* photoreceptors, an observation which was in line with previous data showing the presence of an intact connecting cilium (CC) in the *Prph2^−/−^* photoreceptors [32,33,39]. Additionally, the same study revealed that the Prph2 C-terminus mediates suppression of the formation of ectosomes at the distal part of the CC in rod photoreceptors [33]. Ectosomes are thought to play an important role in the processes of ciliary resorption, regulation of the ciliary protein content and regulation of signal transduction [40,41,42,43,44]. The ability of Prph2 C-terminus to suppress the release of ectosomes offers an explanation to why photoreceptor cilia are able to form the highly complexed intra-ciliary membranous structures like that of the ROS [33]. Further evidence for a key role of the Prph2 C-terminus in the initialization of ROS formation was provided by a recent study that generated a chimeric protein composed of the C-terminus of Prph2 fused to the body of Rom1, the non-glycosylated homologue of Prph2 [34]. Expression of the chimeric protein was sufficient to initialize the formation of ROS structures but failed to elaborate them proving that the Prph2 C-terminus is responsible for the formation of the initial membrane outgrowth while the other parts of Prph2 are required for the development of the highly organized structure of the OS.

*Prph2^+/−^* mice show defects in both function and structure of photoreceptor OS, which proves that the role of Prph2 is not restricted to the initialization of OS formation but also important for its structural development and maintenance [16]. In mature ROS discs, Prph2 is restricted to the rim [15]. In addition, previous studies demonstrated that Prph2 displayed membrane fusion activity, which led to the theory that Prph2 is involved in the shaping of the closed rim of fully developed discs in ROS [45,46]. Indeed, a function of Prph2 in inducing the curvature of the plasma membrane was observed in several studies [25,33,47,48]. While all these studies agree that Prph2 is involved in the generation of membrane curvature, they are contradictory in defining the exact region of Prph2 responsible for this activity. Two studies pinpointed the membrane curvature inducing activity to an α-helix motif located in the C-terminus of Prph2 [47,48], while a third study argues that this motif is rather preventing membrane curvature than inducing it [25]. A recent study found evidence for a function of the tetraspanin core of Prph2 in generating membrane curvature while the C-terminus was unable to perform this function [33]. The localization of Prph2 in newly formed ROS discs provided further evidence for the importance of Prph2 in membrane curvature and disc closure [17]. Newly formed discs in the ROS have a close rim at the site adjacent to the axoneme of the photoreceptor cilium while the opposing side of the disc remains open [49]. In these newly formed discs, Prph2 is restricted to the axonemal closed side of the disc [17]. A recent study confirmed the localization of Prph2 on the axonemal side of newly formed discs in the ROS [50]. An additional finding in this study was that Prph2 is more abundant at the closed rim of newly formed discs than at the rim of mature discs, indicating that Prph2 is not only involved in the closure of this rim but also required for the initialization of the formation of new discs.

The membrane curvature of the open end of newly formed discs in ROS is opposite to the curvature on the closed rim, and thus unlikely to allow the binding of Prph2 [26]. This led to the question of whether a different transmembrane protein may promote the curvature of the open ends. Prominin-1 was identified as a potential candidate involved in the shaping of newly synthetized discs in ROS. Prominin-1 is a membrane protein with five transmembrane domains that localizes at the open end of newly synthesized discs in ROS, thus opposite to the localization of Prph2 [51,52,53,54,55]. The vital role of Prominin-1 in the structural integrity of the photoreceptor OS is supported by the finding that mutations in Prominin-1 are connected to retinal diseases causing degeneration of the photoreceptors [54,56,57,58,59]. Earlier studies on Prominin-1 demonstrated that it shows a binding preference for curved membranes, but does not induce the curvature itself [60,61]. Thus, Prominin-1 is a promising candidate for the maintenance of the curvature on the open end, while the protein responsible for inducing the curvature at these ends remains elusive. A second question, which remains open thus far, is what mechanism restricts Prominin-1 to the newly synthesized discs? When ROS discs mature and reach their final diameter, the open ends close; presumably through the activity of Prph2 and become separated from the plasma membrane of ROS. While Prph2 is located on the entire rim of these closed discs, Prominin-1 is no longer found after the closure of the now matured disc [55].

While rod OS discs are closed and separated from the plasma membrane, cone OSs on the other hand contain open discs also referred to as lamellae and are contiguous with the plasma membrane. Although amphibian COS consists exclusively of open discs, the number of open discs in mammalian COS varies depending on the species [38,62,63,64,65]. The localization of Prph2 in the open discs of COS is comparable with its localization in the newly synthesized ROS discs, with Prph2 being restricted to the side adjacent to the axoneme of the photoreceptor cilium [55]. The discs in COS display a closed rim formation on the axonemal sides. Prominin-1 is localized on the open side, in line with its localization in the newly formed ROS. Studying the role of Prph2 in the development of cones in mice has proven to be difficult due to the low percentage of cone photoreceptors in the murine retina, whereby only ~3–5% of photoreceptors are cones. We have analyzed the effect of loss of Prph2 in the background of an Nrl knockout (*Nrl^−/−^*), in which all rods are converted to S-cone-like cells [66,67,68]. Here the cones in the double knockout mice (*Prph2^−/−^/Nrl^−/−^*) displayed disorganized COSs lacking the flattened lamellae characteristic of WT COS [69]. Furthermore, this study provided evidence for the trafficking of many OS proteins to this altered OS structure, indicating that the transport of those proteins towards the OS does not rely on Prph2 [68]. The ability of *Prph2^−/−^* retinas to form disorganized COSs argues for a different role of Prph2 in cones than in rods. As mentioned above, ROSs fail to develop in the *Prph2^−/−^* mice, thus demonstrating a vital role for Prph2 in the initialization of the formation of this complex intra-ciliary membrane structure in ROS. The discovery of a role for the C-terminal region of Prph2 in inhibiting the release of ectosomes from the rod photoreceptor cilia [33] added new interesting aspect to the process of the rod OS formation. Future studies are needed to determine if this function of Prph2 is preserved in cones.

The identification of Rom1 as a Prph2 interactor initiated a multitude of studies investigating the complex formation between Prph2 and Rom1 [24,26,70]. Like Prph2, Rom1 is a photoreceptor-specific membrane protein and a member of the tetraspanin superfamily [71]. While only sharing a sequence identity of 35% with Prph2, both proteins form a highly conserved and similar secondary and tertiary structures with four transmembrane domains and two intradiscal loops [24]. Prph2 and Rom1 form heteromeric and homomeric tetrameric complexes, held together by non-covalent bonding mediated by the second intradiscal (D2) loop [21,22,23]. These tetramers can assemble into intermediate complexes consisting of both hetero- and homotetramers to establish a mix of covalent and non-covalent linked complexes [72]. These intermediate complexes were found to include at least two tetramers. In addition to the intermediate complexes, higher order complexes containing only Prph2 were also identified [72]. The core of both intermediate and higher order complexes consists of tetramers [21,25]. Interestingly Rom1 is excluded from these higher order complexes [21,25]. How this exclusion is mediated is not fully understood. A possible explanation is that Prph2 can utilize both the conventional secretory pathway through the Golgi apparatus as well as the unconventional pathway which bypasses the Golgi apparatus [26,73]. The presence of Rom1 results in more Prph2 being transported through the conventional pathway [34]. It is likely that this level of sorting during the initial steps of protein trafficking provides a possible mechanism mediating the exclusion of Rom1 from higher order complexes. Studies verifying this mechanism are a possible direction for future work. The higher order complexes are localized at the closed rim of discs indicating an important role for them in the membrane folding necessary to close the disc rim [25,74]. Biochemical evidence supported the idea that during the formation of Prph2/Rom1 complexes, intermolecular disulfide bonding between Prph2 and Rom1 are formed [72,75,76].

The ability of Prph2 and Rom1 to form complexes was proven to be vital for their function in OS formation [18,27,74]. The cysteine at position 150 (C150) in the D2 loop of Prph2 was found to be critical for the formation of higher order Prph2/Rom1 complexes [21,28,77]. C150 of Prph2 forms an intermolecular disulfide bond with C150 of another Prph2 to make homodimers or with C153 of Rom1, thus allowing the formation of Prph2/Rom1 heterodimers and heteromeric tetramers. A knock-in mouse expressing a point mutation at C150 of Prph2 (C150S) provided further evidence for the importance of this cysteine in the formation of higher order complexes and thus for the role of Prph2 and Rom1 in the development of the OS [78]. The formation of intermediate as well as higher order complexes was fully abolished by the C150S mutation and both ROS and COS were severely disorganized. Despite being disorganized, the OS structure of the C150S mice were somewhat better when compared to *Prph2^+/−^* while no functional improvement was observed, indicating that the ability of Prph2 to form intermediate complexes with Rom-1 as well as its ability to form higher order complexes consisting of Prph2 homo-tetramers is not only relevant for maintaining the OS structure but also for its function [78].

The development of higher order complexes seems to be expendable for the initiation of OS formation, since generation of both ROS and COS is initiated in the C150S mutant retinas that are devoid of these complexes. Comparable results were obtained for a mutation deleting lysine at position 153 in the D2 loop of Prph2 (K153Δ) [79]. K153Δ represents a mutation found in patients resulting in variable phenotypes ranging from ADRP to more cone specific MD [80]. This mutation also obliterates the ability of Prph2 to form higher order complexes [79]. The initiation of ROS and COS structures was unaffected while both displayed structural and functional decline. While the C150S and K153Δ mutations prevent the formation of higher order complexes, some pathogenic mutations of *Prph2* lead to the formation of an abnormal high molecular weight aggregates that include both Prph2 and Rom1 [81,82]. A patient mutation causing the exchange of the tyrosine at position 141 for a cysteine (Y141C), effectively adding another cysteine in the D2 loop, causes mostly defects in the macula but some cases of RP were also reported [30,82,83]. The Prph2 and Rom1 complexes in Y141C knockin retinas were significantly altered, showing the formation of abnormal high molecular weight aggregates containing both Prph2 and Rom1. This is a remarkable finding considering that Rom1 is normally excluded from higher order complexes [82]. These mice display structural and functional defects in ROS and COS. The formation of abnormal high molecular weight aggregates could also be found in another patient mutation in which tryptophan in Prph2 is substituted by arginine at position 172 (R172W) [81]. Transgenic mice carrying this mutation displayed the formation of high molecular weight aggregates of Rom1 while Prph2 complexes were not affected. Additionally, these high molecular weight Rom1 aggregates were more abundant in cones, a result that is in line with the decline of cone function in this mouse model [81].

Taken together, these results demonstrate that the exact regulation of the Prph2/Rom1 complex formation is critical for proper OS development, maintenance and disc size in both rods and cones as well as for their function. For OS initiation, proper complex formation seems to be irrelevant. It is worth noting that these studies point to a striking difference between rods and cones with regard to how they are affected by the different *Prph2* mutations. Mutations, which primarily affect the rods, seem to be resulting from haploinsufficiency or loss-of-function effects [78,84,85], while those effecting cones seem to be more susceptible to changes in the complex formation, likely due to gain-of-function defects [78,79,81,82,86,87,88]. Another differentiation between rods and cones is how they process Rom1. While ablation of Prph2 results in the complete loss of ROS and severely disorganized COS, elimination of Rom1 (*Rom1^−/−^*) is less severe, indicating that Rom1’s role is more in fine tuning of disc sizing [27]. It is worth noting that most studies considering these functions were performed in WT murine rods. A recent study performed on a *Nrl^−/−^* background using the K153Δ mutation of Prph2 demonstrated a loss of Prph2/Rom1 interaction specifically in cones [79]. In order to unravel the precise differences in the roles of Rom1 in rods versus cones, further studies are needed.

The D2 loop of Prph2 contains a motif for N-linked glycosylation, a motif which is absent in the D2 loop of Rom1 [24]. In rods, the N-glycosylation on Prph2 was found to be expendable for ROS [89]. In transgenic mice expressing the unglycosylated form of Prph2 specifically in rods, the interaction with Rom1 or the formation of complexes was not affected. Furthermore, expressing unglycosylated Prph2 on a *Prph2^−/−^* background achieved a full rescue of the phenotype, further proving that the N-glycosylation of Prph2 is unnecessary for its function in rods [89]. A knockin mouse model expressing unglycosylated Prph2 [75] recapitulated the rod results observed by Kedzierski et al. [89]. However, while rods were unaffected, cone function was found to decrease significantly in these mice. Furthermore, a significant decrease in Prph2 and Rom1 levels in the cones of the knockin mice was also observed. These results further highlight the differential functional roles of Prph2 in rods versus cones.

Apart from its interaction with Rom1, Prph2 was found to interact with additional OS specific proteins. One of those interactors is with cyclic nucleotide gated channel B1a (CNGB1a) [90,91]. CNGB1a was shown to be relevant for the correct sizing and alignment of the rod discs, supporting the hypothesis that the interaction between Prph2 and CNGB1a plays a role in the proper shaping of the discs [92,93]. Additionally, Prph2 was also found to directly interact with rhodopsin (Rho), thus forming a complex of Prph2, CNGB1a and Rho to anchor the rim of the discs to the OS plasma membrane and the rims of adjacent discs. It has been hypothesized that these interactions are responsible for correct alignment and stacking of the discs [94]. This was supported by findings that co-depletion of these three proteins exacerbated both functional and structural defects in rod photoreceptors [95].

## 3. Insight from Mouse Models into the Pathophysiology of *Prph2* Mutations

### 3.1. Transgenic Mouse Models of Prph2 Mutations

Genetically engineered mouse models expressing pathogenic mutations of Prph2 have proven to be valuable tools in studying the pathophysiology of Prph2 related blinding diseases. These models have been generated as transgenic expressing *Prph2* under the control of heterologous promoters that in some cases are specific to rods, cones or both.

#### 3.1.1. Prph2^R172W^

One of the most common disease-causing mutations of *PRPH2* is the substitution of tryptophan to arginine at position 172 (R172W) [96,97,98,99]. Patients carrying this mutation display a decrease in cone function, while rods’ function remains unaffected [96]. The R172W mutation is located in the D2 loop, which is the region that promotes complex formation and where the vast majority of pathogenic *PRPH2* mutations are located [22,87]. Interestingly, the R172W mutation is located outside the part of the D2 loop involved in the formation of complexes [22]. Expressing the R172W mutation in the presence of the full complement of WT Prph2 led initially to normal rod function and structure [87]. In contrast, the retina of these mice exhibited a significant loss in the number of blue and green cones and associated with a decrease in their photopic responses. When expressed on *Prph2^+/−^* background, the R172W mutation led to a late-onset reduction in the number of rod cells and in scotopic responses, besides the cone phenotype. This varies from the cone exclusive phenotype observed in patients carrying this mutation and is most likely due to haploinsufficiency of Prph2. Expression of the R172W mutation in the *Nrl^−/−^* retina reproduced the functional decline in the photopic responses and provided evidence as to how the R172W mutation disrupted COS structures by causing the formation of abnormal high molecular weight Prph2/Rom1 aggregates [81]. This result demonstrates how the functional decline of the cones occurs in patients carrying this mutation.

#### 3.1.2. Prph2^C214S^

The substitution of a cysteine at position 214 in Prph2 with a serine (C214S) was shown to cause ADRP, a degeneration of rods followed by a late-onset degeneration of cones [100]. Expression of the C214S transgene in the presence of the full complement of WT Prph2 mice failed to produce any phenotype, proving that the phenotype observed in patients is through a mechanism other than dominant gain-of-function [85]. However, when expressed in *Prph2^+/−^* or *Prph2^−/−^* retinas, the C214S protein failed to rescue the functional and structural phenotypes in both rods and cones, indicating that the C214S mutation is resulting in a loss of function. An interesting finding was that the C214S mutant Prph2 was unable to interact with Rom1, indicating an alteration in the formation of complexes caused by this mutation. Although an ample amount of Prph2 C214S transgene message was detected, only a trace amount of the mutant protein was identified, which led to the conclusion of a loss-of-function phenotype associated with the C214S mutation. It is unclear whether the low levels of mutant protein are due to a low rate of synthesis or instability of the mutant protein.

#### 3.1.3. Prph2^P216L^

A substitution of a proline at position 216 in Prph2 with a leucine (P216L) was found to cause RP in patients [101]. Expression of P216L in mice in presence of WT Prph2 (*Prph2^+/+^*) levels led to an age-dependent significant decrease in scotopic responses and shortened rod OSs with normal disc alignments, indicative of a dominant-gain-of-function effect [84]. Moreover, in the presence of one WT allele of *Prph2* (*Prph2^+/−^*), expression of the P216L caused a significant decrease in rod function compared to *Prph2^+/−^* mice. In these mice, a more severe OS shortening and disorganization of the discs could be observed in presence of the P216L mutant protein. Interestingly, expression of the P216L protein in a *Prph2^−/−^* retina caused the formation of small, highly ROS, which represents a limited rescue when compared to the complete absence of ROS in *Prph2^−/−^.* Taken together, these results prove that the P216L mutation in Prph2 results in a dominant-gain-of effect, in agreement with the rod dominant phenotype observed in patients.

#### 3.1.4. Prph2^L185P^

The substitution of leucine at position 185 to proline (L185P) in Prph2 causes a rare digenic form of RP. Patients heterozygous for both the L185P mutation and a null mutation in Rom1 exhibit symptoms of RP, while carriers of any of the two mutations do not show a phenotype [102,103]. Transgenic mice expressing the L185P mutation in a *Prph2^+/−^* or a digenic *Prph2^+/−^/Rom1^+/−^* background displayed a late onset thinning of the outer nuclear layer (ONL) concomitant with reduced scotopic electroretinograms (ERG) [104].

The transgenic mouse models described above provided valuable insights into the pathophysiology of Prph2 related diseases. Since the promoters used to express the transgene are heterologous, often levels of expression of the transgene differ considerably from that of the endogenous leading to a varied phenotype.

### 3.2. Prph2 Knockin Mouse Models

The variation in expression levels of the transgene observed in transgenic mouse models is particularly problematic in cases of reduced levels of the expressed protein since reductions in the expression level of WT Prph2 were shown to result in haploinsufficiency leading to severe retinal defects [16,105]. Haploinsufficiency makes it hard to distinguish between the effects seen in transgenic mice that are due to the mutation or those resulting from the reduced levels of expressed protein. In order to overcome this, recent studies relied on Prph2 knockin mouse models for a set of mutations found in patients of Prph2 related diseases. Below are the models currently presented in the literature and comparison of their retinal phenotypes to patient’s phenotype carrying the same mutation.

#### 3.2.1. Prph2^307/+^ and Prph2^307/307^

A deletion of a single base pair at codon 307 in human *PRPH2* results in a slow progressing form of ADRP [106]. A mouse model in which a targeted single base deletion at codon 307 in *Prph2* was introduced and showed a severe decline in photoreceptor survival and function [107]. ERG revealed a decrease in photoreceptor function in heterozygous animals starting at two months of age, while no ERG responses were detected in the homozygous animals even at one month of age. Retinal phenotypes in the heterozygous or homozygous mice were more severe than in age-matched *Prph2^+/−^* and *Prph2^−/−^* mice, respectively. Thus, the deletion mutation resulted in a strong dominant gain-of-function effect. This knockin model represents a drastic case of a retinal phenotype that differs greatly from that observed in the patients whereby the structural and functional decline is slow, starting in the fifth or sixth decades of life. The mouse model, on the other hand, displays a rapid degeneration with an early onset. The deletion at codon 307 in the human *PRPH2* gene causes a frameshift and creates a stop codon after the addition of 16 amino acids. If the resulting mutant protein is stable, it is expected to be 26 amino acid shorter than the wildtype. It is likely the case since the phenotype seen in patients is mild due to gain-of-function defect, but if the mutant protein is unstable, then the phenotype would be loss-of-function defect similar to that of haploinsufficiency. However, in the mouse genome, such one base deletion at codon 307 is predicted to result in the alteration of the last 40 amino acids of the C terminus of the protein and the addition of 11 extra amino acids. This results in the translation of 51 amino acids after codon 307 in the mouse [107]. The different effects on the translated protein caused by the deletion in codon 307 observed between human and mouse explains the differences in the severity of the retinal phenotypes. It is important to note that retinal phenotype in the homozygous mouse is reported to be worse than the complete-loss-of-function phenotype seen in the Prph2 null mice. This observation suggests that the mouse phenotype is likely a combination of loss- and gain-of-function defects and that the toxicity beyond that seen in the Prph2 knockout is probably the outcome of the latter. Since the mouse model was not assessed for the presence of the mutant protein, at this point, it is unclear whether the toxicity arises from the absence of the protein, the 51 altered amino acids at the C-terminus or the lack of endogenous 40 amino acids at the C-terminus. Obviously, the addition of 16 altered amino acids in the human PRPH2 and shortening the protein by 26 amino acids at the C-terminus have less drastic effect on the retina than those modifications occurred in the mouse protein.

#### 3.2.2. Prph2^C213Y/+^ and Prph2^C213Y/C213Y^

The C213Y mutation is located in the D2 loop of Prph2, in the motif that mediates intramolecular and intermolecular disulfide bonds, thus responsible for the formation of Prph2/Prph2 and Prph2/Rom1 tetramers as well as for the formation of intermediate and higher order complexes [77]. The heterozygous mice (*Prph2^C213Y/+^*) displayed a shortened and disorganized ROS. When compared to *Prph2^+/−^* mice, the ROS of *Prph2^C213Y/+^* mice showed a slight improvement in the stacking and alignment of the discs (Figure 1A,B) [76].

This is evident from the well stacked OS discs seen in some photoreceptors while others looked like whorls similar to those seen in the *Prph2^+/−^* (arrows in Figure 1). Scotopic ERG responses of the *Prph2^C213Y/+^* mice were significantly reduced starting at P30 (Figure 2) and persist all the way to P365. The scotopic response was nearly completely abolished in *Prph2^C213Y/C213Y^* animals and the OS was almost non-existent. The photopic responses were significantly decreased in both heterozygous and homozygous animals at P30. At later time points, the photopic response decreased further in the *Prph2^C213Y/+^* mice and was completely absent in the *Prph2^C213Y/C213Y^* animals. A key finding in this study was that the *Prph2^C213Y/+^* mice, that represent the genotype present in patients, showed better retinal structure despite the fact that both of rod and cone ERG responses were reduced when compared to *Prph2^+/−^* animals.

A key requirement for the function of Prph2 is its ability to interact with Rom1 and to form oligomeric complexes [18,27,74]. Co-immunoprecipitation (co-IP) experiments using retinal lysate from *Prph2^C213Y/C213Y^* mice revealed the inability of Prph2^C213Y^ to interact with Rom1. The homomeric interaction of Prph2 was also reduced in these mice as evident from reduced ability of Prph2^C213Y^ to form intermediate and higher order complexes as determined by sucrose gradient velocity sedimentation [76]. The reduction in oligomeric complexes offers an explanation for the functional decline observed in both *Prph2^C213Y/+^* and *Prph2^C213Y/C213Y^* mice and no abnormal high molecular weight aggregates were observed under non-reducing conditions. *Prph2^C213Y/+^* retina retains some Prph2 in the IS, shown by immunofluorescence (IF) staining of Prph2 (green and arrows in Figure 3) and IS maker syntaxin 3B (STX3B) (red, Figure 3). However, *Prph2^C213Y/C213Y^* lacks the ability to form complexes which leads to complete retention of Prph2 in the inner segments (IS), and perinuclear region while a smaller amount of Rom1 was retained in the IS [76].

Patients carrying the C213Y mutation in *PRPH2* display a butterfly-shaped pattern/macular dystrophy, while a rod dominant RP phenotype is absent [29,108,109,110]. The knockin mouse model for the C213Y displays functional defects in both rods and cones. While the defects in the cones are in line with the defects observed in the cone rich macula of the patients, the defects in the rods displayed by the model are not. The difference between phenotypes in patients and the mouse model may be due to the fact that the mouse retina consists almost exclusively of rod photoreceptors. Since the murine retina lacks a macula, it was not possible to reproduce the butterfly shaped macular dystrophy in the mouse model. One interesting finding in the Prph2^C213Y^ mouse model was the observation of a yellow flecking in the fundus of both *Prph2^C213Y/+^* and *Prph2^C213Y/C213Y^* mice at P180. This phenotype mimics funduscopic anomalies found in patients carrying the C213Y mutation [110].

Gene supplementation was performed by crossing a WT Prph2 overexpressing mouse line (NMP) onto hetero- (*Prph2^C213Y/+^*) or homozygous (*Prph2^C213Y/C213Y^*) mice. The defects observed in the protein trafficking as well as in OS structure were rescued in *Prph2^C213Y^* mice. A rescue on the functional level however could not be observed [76]. These results show that the presence of the mutant protein has a detrimental effect on photoreceptor function.

#### 3.2.3. Prph2^Y141C/+^ and Prph2^Y141C/Y141C^

Like the C213Y mutation, the Y141C mutation of Prph2 is also located in the D2 loop. The OSs in the Y141C knockin mouse model (*Prph2^Y141C/+^* and *Prph2^Y141C/Y141C^* for heterozygous and homozygous animals, respectively) displayed structural anomalies. In *Prph2^Y141C/+^* mice, the OS was shortened with some structural alterations of the discs, including lengthening and vesicular structures [82]. When compared to the *Prph2^+/−^* mice, the structure was better conserved in the *Prph2^Y141C/+^* retina (Figure 1B). In the *Prph2^Y141C/Y141C^* mice, OS formation is initialized but neither mature discs nor lamellae could be observed. Instead, the OS contained flattened whorl shaped membranous structures with vesicular arrangements lining up adjacent to them. Scotopic ERG revealed a significant decrease in rod function displayed by the *Prph2^Y141C/+^* mice at P30 (Figure 2). Interestingly, the scotopic ERG did not deteriorate further at P180 [82].

Non-reducing SDS-PAGE/immunoblots of *Prph2^Y141C/+^* retinal lysates showed abnormal high molecular weight aggregates at 250kDA. This was further evaluated by sucrose gradient velocity sedimentation experiments which revealed the formation of the expected intermediate and higher order complexes with an aberrant high molecular weight band indicating that the Y141C mutation results in the formation of abnormal large aggregates [82]. In the *Prph2^Y141C/Y141C^* retinas, the observed phenotype was even more pronounced with an increase in the aberrant high molecular weight aggregates at the expense of the formation of the normal intermediate and higher order complexes. While in WT mice Rom1 is normally excluded from higher order complexes, in the mutant retina, a portion of Rom1 was incorporated in the aberrant high molecular weight aggregates. Surprisingly though, mutant Prph2 was transported correctly to the OS which is shown by IF staining of Prph2 (green) and IS marker STX3B (red) (Figure 3). However, the formation of these high molecular weight aggregates likely interfered with the normal function of Prph2 and likely Rom1 in the OS. This offers an explanation for the structural and functional phenotypes observed in the knockin mice.

Patients carrying the Y141C mutation in *PRPH2* display primarily defects in the macula with some reported cases of more rod-associated phenotypes such as night blindness and RP [30,82,83]. The *Prph2^Y141C/+^* mice, which represent patients’ genotype, exhibit a decline in rods’ function, mimicking the observed rod phenotype in some patients. Again, the difference between the knockin model and the patient’s phenotypes can be seen in the cone function. The most prominent phenotype found in patients with the Y141C mutation is the functional and structural decline in cones. The *Prph2^Y141C/+^* mice on the other hand displayed only a slight functional decline in the photopic ERG which was not found to be significant [82], indicating that the cone function in these mice is not severely affected. While the heterozygous mice do not show significant decline in cone function, they do display a flecking in funduscopic analyses [82], which mimics findings in the fundus of patients. Recently, a study was undertaken to identify a potential explanation for the huge variation in phenotypes observed in patients by determining the role Rom1 plays in the observed phenotype [88]. Mice heterozygous or homozygous for the Y141C mutation were crossed with Rom1 knockout mice to produce mice that express mutant Prph2 in absence of Rom1 (*Prph2^Y141C/+^/Rom1^−/−^* and *Prph2^Y141C/Y141C^/Rom1^−/−^*). Absence of Rom1 abolished the formation of the abnormal high molecular weight aggregates and accumulation of mutant Prph2 in the IS and ONL in *Prph2^Y141C/Y141C^/Rom1^−/−^* retinas [88]. The depletion of Rom1 also changed the symptoms seen in the Y141C model. While the photopic ERG amplitudes in *Prph2^Y141C/+^/Rom1^−/−^* mice were comparable to that of the WT, a significant reduction in the scotopic ERG responses were observed [88].

When compared to the cone-rod functional defects noted for the *Prph2^Y141C/+^* mice, *Prph2^Y141C/+^/Rom1^−/−^* mice mainly displayed a rod-dominant functional defect. In addition, the funduscopic anomalies were almost completely abolished in the *Prph2^Y141C/+^/Rom1^−/−^* mice. The results obtained in this study prove that alteration in the level of Rom1 can change the phenotype caused by a pathogenic *Prph2* mutation, and thus potentially provide an explanation to the variable phenotypes seen in patients carrying the same *PRPH2* mutation. Further studies combining *Prph2* mutations and mutations in *Rom1* should provide further insights into the complexity of phenotypes seen in the various *PRPH2* related diseases.

#### 3.2.4. Prph2^K153Δ/+^ and Prph2^K153Δ/K153Δ^

Another mutation in *PRPH2* that leads to variable phenotypes among patients is the deletion of codon 153 (K153Δ) that results in the elimination of the lysine at position 153 in the D2 loop of Prph2. This mutation is found to associate with RP, pattern dystrophy and fundus flavimaculatus [80]. K153Δ-Prph2 knockin mouse model was generated and provided evidence that the mutant protein cannot form the complexes required for OS formation [79]. The heterozygous knockin mice (*Prph2^K153Δ/+^*) displayed a shortened OS with minor structural defects at P30 [79] (Figure 1). At P180, these animals also exhibited a reduction in ONL thickness without further structural deterioration of the OS [79]. The overall Prph2 protein level in these mice was around 80% compared to the level in WT mice, thus demonstrating that the observed structural defects are most likely caused by the dominant effect of the mutant protein rather than due to haploinsufficiency.

The heterozygous animals displayed a significant progressive reduction in scotopic responses that started as early as P30 (Figure 2) and worsened with age. The photopic response in the *Prph2^K153Δ/+^* was also significantly reduced at P30, albeit this reduction did not worsen as the animals aged (Figure 2). In the homozygous animals, the scotopic and photopic responses were minimal at P30. Biochemical analysis showed that the formation of covalently linked Prph2 dimers in *Prph2^K153Δ/K153Δ^* mice was abolished while Rom1 homodimers were present [79].

Due to the observed effects on cone function, *Prph2^K153Δ/+^* animals were crossed into the *Nrl^−/−^* background in order to assess the effects of the K153Δ mutation on cones. Photoreceptor cells in the resulting *Prph2^K153Δ/+^/Nrl^−/−^* retina displayed a highly disrupted structure with many photoreceptors having no lamellae what so ever. Interestingly, the mutant Prph2 was able to interact with Rom1 in the rod-dominant *Prph2^K153Δ/K153Δ^* retinas, but this interaction was abolished in *Prph2^K153Δ/K153Δ^/Nrl^−/−^* retina, indicating a defect in the Prph2/Rom1 interaction specific to cones [79]. Sucrose gradient velocity sedimentation showed no significant alteration in complex formation in *Prph2^K153Δ/+^* retina when compared to WT while in the *Prph2^K153Δ/K153Δ^* retina, the formation of intermediate and higher order complexes is abolished [79]. Here, both Prph2 and Rom1 are restricted to the tetramer fractions. The same was observed in the *Prph2^K153Δ/K153Δ^/Nrl^−/−^* retinas. Sedimentation profile showed that the amount of higher order complexes in the *Prph2^K153Δ/+^/Nrl^−/−^* retina was reduced, while unaffected in *Prph2^K153Δ/+^* retina [79]. This provides further evidence for a differential role of the lysine at position 153 in rods and cones, and hence emphasizes the notion for potential varied roles for Prph2 in rods versus cones. Localization studies performed in *Prph2^K153Δ/+^* mice demonstrated that most of Prph2 and Rom1 were successfully transported to the OS with some amount of Prph2 mislocalized to the IS (arrows in Figure 3). Small amount of rhodopsin (Rho) and M-opsin were also found to be mislocalized to the ONL and outer plexiform layer (OPL) in this model [79].

In the *Prph2^K153Δ/K153Δ^* retinas, the majority of Rho and Prph2, but not Rom1, were found to be mislocalized to the ONL. As stated above, patients with the K153Δ mutation exhibit a highly variable phenotype, ranging from rod dominant RP to more cone related defects in the macula [80]. The K153Δ knockin mouse model displayed functional and structural defects in both rods and cones, and thus mimics the phenotype seen in patients carrying this mutation. While the lack of a macula in the murine retina made it impossible to observe the macular pattern dystrophy often found in patients [80], the knockin mouse still showed funduscopic anomalies [79] which are characteristic of the pattern dystrophy in patients. *Prph2^K153Δ/+^* mice show a flecking in the fundus at P180 which is more severe in the *Prph2^K153Δ/K153Δ^* mice. This phenotype does not deteriorate further in P365 heterozygous animals while the flecking was replaced by large splotches in the homozygous animals [79].

Gene supplementation using the NMP mouse that over-express Prph2 (*NMP/Prph2^K153Δ/+^* and *NMP/Prph2^K153Δ/K153Δ^*) rescued the structural defects but failed to rescue the functional decline seen in scotopic and photopic ERGs [79]. This indicates that the presence of the mutant protein alone is sufficient to deteriorate the photoreceptor function and suggest that gene silencing along with gene augmentation is the best strategy for this model.

The knockin models for *PRPH2* related patient mutations proved to be very useful in highlighting dominant-effects of the *Prph2* mutations. In general, the models were more successful in mimicking patient phenotypes related to a decline in rod function. Patient phenotypes related to functional defects in the cones or pattern dystrophies in the macula were more difficult to reproduce in mice due to the lack of a macula and a lower overall percentage of cones in the murine retina. Crossing the knockin mouse models with *Nrl^−/−^* mice, as done in the studies with the K153Δ model [79], has proven to be a successful approach in studying the effects of the mutation on cones in detail. While the knockin models could not reproduce the macular pattern dystrophy, they successfully reproduced the funduscopic aberrations, which connect with the pattern dystrophy.

#### 3.2.5. Prph2^N229S/+^ and Prph2^N229S/N229^

The knockin mouse that alters the N-linked glycosylation at asparagine 229 (N229S) in the D2 loop was used to study if Prph2 glycosylation plays a role in its interaction with Rom1. Heterozygous mice (*Prph2^N229S/+^)* did not have any significant changes in structure or function of the OS [75]. However, homozygous mice (*Prph2^N229S/N229S^*) displayed a late onset thinning of the outer nuclear layer (ONL) and occasional abnormal disc staking in cones and a slightly reduced photopic ERG at P180. Since Prph2 could not be glycosylated, higher order complexes were decreased and there was an increase of Prph2 and Rom-1 in the intermediate complexes [75]. Therefore, it was concluded that the glycosylation plays a major part in regulating the interaction between Prph2 and Rom1, which is critical for cone health.

Table 1 and Table 2 summarize the phenotypes associated with the Prph2 knockin models, highlighting rod and cone structure, Scotopic and Photopic ERG, complex formation and protein localization. It also summarizes fundus observations, and patient’s phenotypes whether mainly rod- or cone-specific defects or the combination of the two. Figure 1 and Figure 2 show structural and functional differences between Prph2 heterozygous knockin mutations, while Figure 3 is a representative IF showing retinal localization of Prph2 among the models.

## 4. Gene Therapy of *PRPH2* Mutations

There is a vast variety of *PRPH2* mutations associated with autosomal dominant retinal degenerative diseases, including RP, several forms of macular dystrophy and cone rod dystrophies [29]. Gene therapy seems to be a promising approach in the treatment of those *PRPH2* associated diseases. The small size of the *Prph2* cDNA, which is approximately 1.1 kb, is advantageous since many vectors used for gene therapy can only carry DNA of a limited size. Despite the variety of feasible approaches available for the gene therapy of *Prph2*, thus far no treatment ready for clinical trials has been developed.

The expression of the WT Prph2 in the background of pathogenic *Prph2* mutations represents one of the most promising approaches. Pioneer studies using transgenic mice expressing WT Prph2 under different promoters in a *Prph2^+/−^* or *Prph2^−/−^* background revealed a significant increase in both structural and functional properties of rods and cones [105,112]. Replacing Prph2 in *Prph2^−/−^* mice utilizing adeno-associated virus (AAV) carrying *Prph2* regulated under the rhodopsin promoter provided promising results [113]. Here the subretinal injection of the AAV resulted in a partial rescue of ROS structure as well as scotopic ERG response. A follow up study was able to show that repeated injections with the AAV resulted in an even more pronounced rescue of the phenotype with an increase in the scotopic b-wave response observed in the injected mice [114]. In this study, several time points after the injections were analyzed in order to validate whether the observed rescue was long lasting. This analyses revealed that the rescue achieved by the AAV injection was lost after 15 weeks post injection [114]. In addition to the loss of the rescue with time, the amplitude of the scotopic b-wave was significantly lower than in WT mice and the scotopic a-wave was not improved by the injection of the AAV. A reason for this is that the injection of the AAV in mice only resulted in a low transduction rate of roughly 30% [114]. An improvement in the transduction rate will thus be necessary in order to develop a viable AAV based gene therapy.

Nanoparticles (NP) represent a second promising approach in transferring WT Prph2. These particles were found to be well tolerated by the retina, even after multiple injections, and have a high DNA capacity up to 14 kb, as tested in the eye [115,116,117,118,119,120]. NP carrying full-length murine *Prph2* cDNA under the control of either rod or cone specific IRBP promoters or ubiquitous chicken beta actin promoter was injected in the retina of *Prph2^+/−^* mice [121]. The injection resulted in a partial rescue of OS structure and also prevented the thinning of the ONL. These rescue effects lasted 15 months post injection and were most pronounced near the site of injection [121]. While the NP treatment could overcome the decline in rescue overtime, the effect on the photoreceptor function remained limited. NP injection resulted in a small yet insignificant improvement in the scotopic a-wave but a significant improvement in the photopic b-wave. The results obtained with the different promoters used were comparable. The small benefit of the NP injection observed in the functional tests might be due to an incomplete distribution of the NPs in the retina. In line with this, the structural improvements observed after NP injection were best close to the site of injection. Improving on the distribution and uptake of the used vector in the retina, regardless if AAVs or NPs are used, seems to be a necessary next step in order to achieve a more pronounced functional rescue.

The studies above described the replacement of Prph2 in either *Prph2^+/−^* or *Prph2^−/−^* mice, thus in a scenario whereby Prph2 is either absent or haploinsufficient. However, most patients suffer from a dominant mutation in *PRPH2*. In order to analyze the efficiency of treatment in a scenario closer to actual *PRPH2* related diseases, the knockin mouse models carrying a disease related *Prph2* mutation were analyzed. Both *Prph2^K153Δ/+^* and *Prph2^C213Y/+^* mice were crossed with a normal-Prph2-overexpressing mouse line (NMP) [76,79]. The resulting *Prph2^K153Δ/+^/NMP^+/−^* and *Prph2^C213Y/+^/NMP^+/−^* mice displayed a rescue in OS structure, protein expression levels and trafficking, but no rescue in both rod and cone functions [76]. In both cases, the presence of the mutant protein continues to affect the photoreceptor function. This is due to dominant-gain-of-function effects caused by these mutations. These results show that, in addition to gene augmentation therapy, silencing the mutant *Prph2* allele is essential for ultimate rescue. A combined approach of short hairpin (sh)-RNA mediated knockdown and expression of a shRNA resistant protein has been performed in models for ADRP carrying pathogenic mutations of rhodopsin [122,123]. Here, both WT and mutant Rhodopsin were knocked down by sh-RNA carried in AAV together with a sh-RNA resistant rhodopsin. The expression of the sh-RNA resistant rhodopsin following the knockdown rescued the phenotype. Studies combining the knockdown of mutant Prph2 with the expression of WT Prph2 provided first promising results thus far showing partial functional and structural rescue in mice expressing pathogenic mutations of Prph2 [124,125].

The examples above show that while progress was made in the gene therapy of *PRPH2* related diseases, there are many factors, which need to be considered for the development of a successful therapy. Further complications might arise from the fact that the functional role of Prph2 seems to vary in rods and cones. In addition to that, secondary effects of *PRPH2* mutations on the RPE and the choroid could be observed [29,30,31]. A high variability in the clinical phenotypes displayed by patients, even when carrying the same mutation, represents another challenge, which has to be overcome in order to treat *PRPH2* related diseases.

## 5. Conclusions and Perspectives

Prph2 plays a key role in the maintenance as well as the development of photoreceptor OS. While Prph2 is found to be vital for both rods and cones, there seem to be distinct differences in its function in these two types of photoreceptor cells. A complete knockout of Prph2 results in the complete absence of ROS while COS formation is still initialized even though the resulting COS are severely disorganized and lack lamellae and discs. This begs the question, which protein is mediating the formation of the COS and if it works in an interplay with Prph2. The OS of both rods and cones represent a highly modified primary cilium. The Prph2 mediated suppression of ectosomes shedding from the photoreceptor cilium was found to be a prerequisite for the formation of the ROS. Thus far there is no study demonstrating a similar mechanism in the formation of COS. The investigation of the formation of ectosomes and membrane dynamics in the development of COS in WT and *Prph2^−/−^* mice might be a potential approach for further studies aiming to unravel the different function of Prph2 in ROS versus COS. While the function of Prph2 in the initialization of ROS and COS varies, it seems to be indispensable for the correct shaping, sizing and stacking of discs and lamellae in both. Furthermore, not only the presence of Prph2 is needed for developing and maintaining these structures but also the precise regulation in the formation of the different Prph2/Rom1 and Prph2/Prph2 complexes. Changes in the ratio of the tetramers, intermediate complexes and higher order complexes were proven detrimental to the function as well as the structure of the photoreceptors. The transgenic and knockin mouse models discussed here proved the importance of the precise regulation of the complex formation. Each of the knockin mouse models carrying a pathogenic *Prph2* mutation displayed an altered ratio of the different Prph2/Rom1 and Prph2 complexes resulting in disorganized OS structure and decreased response found in ERG measurements. Interestingly rods and cones were differently affected by the alterations in complex formation, providing further evidence for the differential use of Prph2 by the two photoreceptor cell types. In addition to that, the mouse models used helped identify whether a mutation results in a loss-of-function or a gain-of-function effect.

While there is agreement in the numerous studies concerning the ratio between tetramers, intermediate and higher order complexes and its vital importance for OS structure and function, the precise roles of the different complexes are still not fully understood. In the C150S knockin mouse model, both Prph2 and Rom1 are found exclusively as tetramers, while intermediate and higher order complexes are absent. Still the formation of both ROS and COS is initiated, proving that the tetramers alone are sufficient to initiate OS formation. Homozygous animals fail to form disc and lamellae, indicating that the intermediate and higher order complexes are more likely to be involved in the formation of the rim, membrane closure and disc stacking. The knockout of Rom1 showed minor effects on disc alignment, sizing and stacking while the disc rim formation is unaffected. Rom1 is found in non-covalent hetero-tetramers and in intermediate complexes but it is excluded from higher order Prph2 complexes. It seems plausible that the Prph2 higher order complexes, which are unaffected in the Rom1 knockout retinas, mediate the membrane curvature and rim closure, while the intermediate hetero complexes support disc spacing, sizing and alignment. Additional studies targeting the formation of the different complexes specifically and analyzing their function in rods and cones are needed to further pinpoint the roles of the different Prph2 and Rom1complexes.

Progress in the development of a feasible gene therapy of Prph2 related diseases has been made in recent years. Transducing the photoreceptors of *Prph2^−/−^* and *Prph2^+/−^* mice either via AAVs or NPs showed a partial rescue of the knockout phenotype, even though the magnitude of the rescue and (in case of the AAVs) the persistence of the rescue effect is still low. The small magnitude of the rescue is most likely due to a low transduction rate of photoreceptors by the vectors used. Improving the transduction rate achieved by the vector as well as its distribution in the eye is one obstacle which needs to be overcome in future attempts. A second obstacle is that pathogenic *PRPH2* mutations do not result in a knockout scenario, but instead in a scenario where the mutant protein is still expressed. The knockin mouse models summarized in this review have proven to be a valid approach in understanding how the presence of the mutant protein affects both rod and cone photoreceptors, and also how stable the mutant protein is. In addition, these models helped to identify mutations, which result in dominant effects. Using a sh-RNA-mediated knockdown of both WT and mutant proteins followed by the expression of a sh-RNA resistant form of the WT protein provided a promising approach in the therapy of dominant gain-of-function mutations of Rho and Prph2. The fact that rods and cones utilize Prph2 differently could lead to further complications; this issue needs to be taken into consideration when developing effective therapeutic strategies for *PRPH2* related diseases. Testing these strategies in the murine models adds another layer of complication due to the low percentage of cones, which makes it difficult to address cone-dominant phenotypes in mouse models. Analyzing *Prph2* mutations in the *Nrl^−/−^* background provides an approach for studying functional and structural effects of mutation specifically on cones. The lack of a macula in the murine retina continues to be the rate limiting factor which prevents conclusive studies aimed at developing therapies for mutations associated with macular defects and pattern dystrophy using mouse models. In addition to this, several *PRPH2* mutations were found to cause secondary effects in the choroid and RPE. The reasons for these secondary defects as well as their impact on the pathogenesis of *PRPH2* associated diseases are not well understood. Characterizing these secondary defects has to be achieved in future studies in order to devise a successful therapy.

## Figures and Tables

**Figure 1 cells-09-00784-f001:**
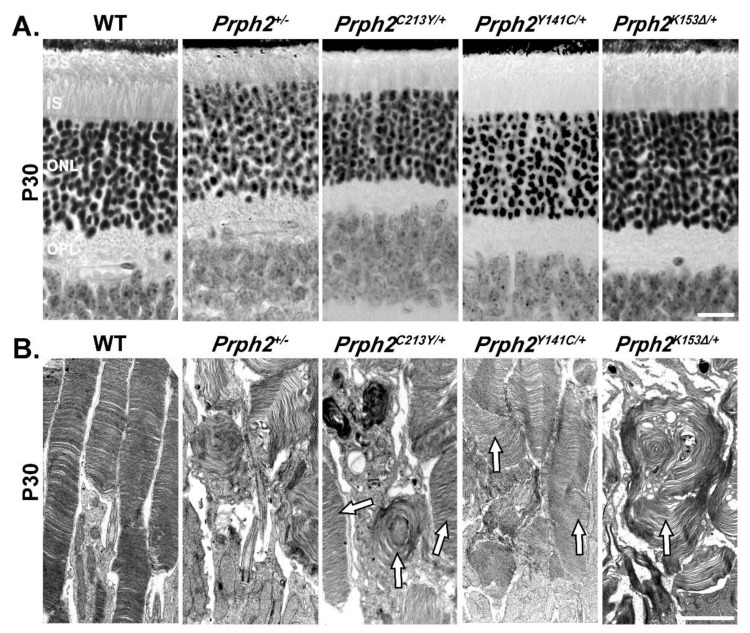
Mutations in the mouse *Prph2* gene lead to varying degrees of photoreceptor degeneration. (**A**) Representative light microscopic images from hematoxylin and eosin stained retinal sections at P30 aligned at the upper edge of the retinal pigment epithelium (RPE). (**B**) Transmission electron microscopic (TEM) images of the interface between the IS and OS of photoreceptors of the indicated genotypes. OS, outer segments; IS, inner segment; ONL, outer nuclear layer; OPL, outer plexiform layer. Scale bars: 20 µm for A and 2 µm for B. Eyes used in this study were dissected, fixed and embedded as previously described [79]. Images were captured at 40× and converted to black and white using ZEN Image Analysis software. The plastic-embedding and TEM methods were as described previously [85]. Images were adjusted and cropped using Adobe Photoshop CS5.

**Figure 2 cells-09-00784-f002:**
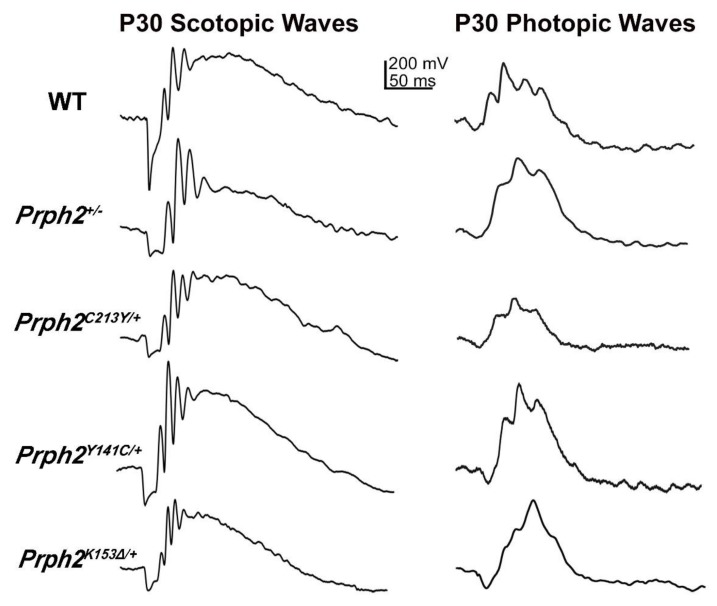
Mutations in *Prph2* hinder OS function assessed via scotopic and photopic electroretinograms (ERGs) at P30. Full-field ERGs were recorded under scotopic and photopic conditions. Shown are representative ERG waveforms from the indicated genotypes at P30. Full-field ERG tests were performed as previously described [85]. After overnight dark adaptation, mice were anesthetized and their pupils dilated. ERGs were recorded with a UTAS system (LKC, Gaithersburg, MD, USA). Waveforms were exported into GraphPad software to obtain wave traces and then exported into Photoshop using a uniform scale.

**Figure 3 cells-09-00784-f003:**
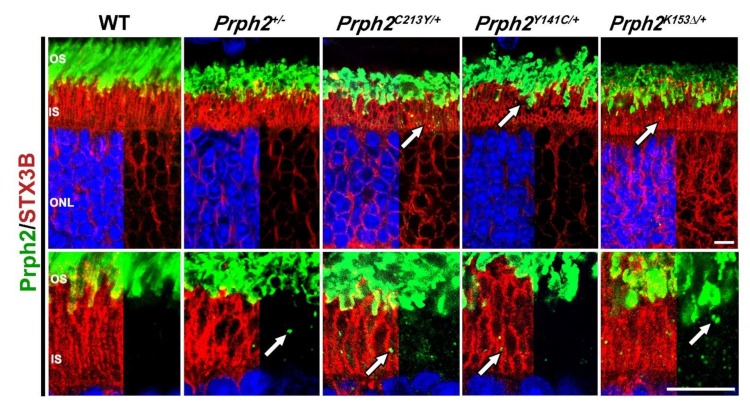
Mutated Prph2 protein traffics to the OS while a small pool is retained in the inner segment. Retinal sections at P30 from the indicated genotypes were probed with antibodies against Prph2 (green) and syntaxin 3B (STX3B) (red). Arrows indicate regions of mislocalization of Prph2. OS, outer segments; IS, inner segment; ONL, outer nuclear layer. Scale bar: 20 µm. Primary antibodies used for immunostaining were polyclonal antibody against Prph2 C-terminus (Prph2-CT) [22] and monoclonal antibody against STX3B [111] (inner segment marker) diluted at (1:1000). AlexaFluor conjugated secondary antibodies (Alexa 488 Rabbit and Alexa 555 Mouse, Life Technologies/ThermoFisher) were used at a dilution of 1:1000 for 2 hours at room temperature. Images were captured on a ZEISS Confocal LSM 900 microscope equipped with a Zeiss Axiocam (Zeiss, Jena, Germany) using a 63× (oil, 1.4 NA) objective. Images were then processed using ZEN Image Analysis software (Zeiss, Jena, Germany). All images shown are orthogonally projected from an eight slice confocal z-stack.

**Table 1 cells-09-00784-t001:** *Prph2* mutations and correlating phenotypes.

Genotype	*Prph2^+/−^*	*Prph2^−/−^*	*Prph2^K153∆/+^*	*Prph2^K153∆/K153∆^*	*Prph2^Y141C/+^*	*Prph2^Y141C/Y141C^*
**Mouse Rod Structure**	Short OSs with whorl structures and ONL thinning at P180.	No OS structures.	Short OSs with whorl structures and ONL thinning at P180.	Almost no Oss with rare whorl structures and ONL thinning at P30, more severe at P180.	Short OS with longer discs and accumulation of vesicular structures, no ONL thinning at P30, ONL thinning at P180.	Small OS with flattened whorls and vesicular structures at P30, no OS present at P180, ONL thinning at P30, more severe at P180.
**Mouse Cone Structure**	Oss with whorl structures	Open OS with no lamella	Occasional whorl shaped OS, mostly open OS with no lamella	Occasional COS seen but mostly absent.	Abnormal and short COS structures.	Occasional COS seen but mostly absent.
**Scotopic ERG**	57% and 33% reduction in a- and b-wave at P30. 74% and 48% reduction in a- and b-waves at P180, respectively.	96% and 93% reduction in a- and b-wave at P30, respectively.	63% and 49% reduction in a- and b-wave at P30. 83% and at 60% reduction in a- and b-wave at P180, respectively.	90% and 89% reduction in a- and b-wave at P30, respectively.	54% and 27% reduction in a- and b-wave at P30 50% and 25% reduction in a- and b-wave at P180, respectively.	90% and 78% reduction in a- and b-wave at P30. 95% and 94% reduction in a- and b-wave at P180, respectively.
**Photopic ERG**	Photopic b-wave comparable to WT at P30. 35% reduction in photopic b-wave at P180.	91% reduction in b-wave at P30.	24% reduction in b-wave P30 and 50% at P180.	64% reduction in b-wave at P30.	10% reduction in b-wave at P30 and P180.	64% reduction in b-wave at P30 and 90% at P180.
**Complex formation**	Prph2 complexes and distribution unchanged, 50% less Prph2 and Rom1. On NR^L-/-^ background, higher order complexes decreased.	No Prph2 present, Rom1 still present but at lesser amount.	Prph2 complexes and distribution unchanged, while Rom1 shifted towards tetramers. On NRL^−/−^ background, higher order complexes decreased, shift of Prph2 and Rom1 towards intermediate complexes.	No Prph2 dimers were formed, while Rom1 dimers were still formed. Prph2 interacted with Rom1. Prph2 and Rom1 restricted to tetramers. On NRL^−/−^ background, no Prph2 dimers were formed, while Rom1 dimers were still formed. Prph2 did not interact with Rom1. Prph2 and Rom1 restricted to tetramers.	Prph2 occasionally found in abnormal high molecular weight aggregates. Abnormal aggregates were held together by intermolecular disulfide bonds. Rom1 also present in abnormal aggregates. Intermediate and higher order complexes formed but abnormal high molecular weight aggregates also present.	Prph2 almost exclusively found in abnormal high molecular weight aggregates. Abnormal aggregates were held together by intermolecular disulfide bonds. Rom1 also present in abnormal aggregates. Intermediate and higher order complexes reduced in favor of the abnormal high molecular weight aggregates.
**Protein localization**	Some rhodopsin detected in the IS and ONL.	Rhodopsin mislocalized to IS and ONL.	Small amount of rhodopsin and M-opsin mislocalized in the IS and ONL.	Huge amount of rhodopsin and Prph2 mislocalized in the IS and ONL.	NA	NA
**Fundus**	No abnormality	Flecking and splotches at P360 and older.	Flecking at P180 and no change at P365.	Severe flecking at P180 and big splotches at P365.	Flecking at P180.	Flecking at P180.
**Rod defect in patients**	NA	NA	RP	NA	Night blindness and RP reported in some patients.	NA
**Cone defect in patients**	NA	NA	Pattern dystrophy and fundus flavimaculs.	NA	Pattern dystrophy changed fundus in macula.	NA
**Reference**	[26,76,79]	[26,76,79]	[79]	[79]	[82]	[82]

**Table 2 cells-09-00784-t002:** *Prph2* mutations and correlating phenotypes (*continued*).

Genotype	*Prph2^C213Y/+^*	*Prph2^C213Y/C213Y^*	*Prph2^N229S/+^*	*Prph2^N229S/N229S^*	*Prph2^C150S/+^*	*Prph2^C150S/C150S^*	*Prph2^307/+^*	*Prph2^307307^*	
**Rod structure mouse**	Shortened OS, irregular structure, some disc structure better organized than in *Prph2^+/−^.* ONL thinning at P30.	Short OS formed with highly disorganized discs. Severe ONL thinning at P30.	Structure unaffected.	Modest ONL thinning at P180.	Modest elongation of discs, occasional formation of whorls. No ONL thinning.	Shortened OS, elongated discs curving into whorls. No ONL thinning.	ONL thinning starting at P60. Rod OS shortened, formation of whorls.		ONL thinning starting at P30 and rod OSs and absent.
**Cone structure mouse**	Well organized lamella at P30 but slightly shorter.	Abnormally stacked lamella with whorl shaped structures.	Normal COS structures.	Occasional abnormal disc stacking at P180	NA	Shortened OS, elongated discs curving into whorls.	NA		Cone OS are absent.
**Scotopic ERG**	60% and 47% reduction in a- and b-wave P30. 89% and 67% reduction in a- and b-wave at P180, respectively.	96% and 94% reduction in a- and b-wave P30, respectively.	ERG a- and b-wave comparable to WT at P30 and P180.	ERG a- and b-wave comparable to WT at P30 and P180.	50% and 30% reduction in a- and b-wave at P30, respectively.	75% and 56% reduction in a- and b-wave P30, respectively.	60% and 62% reduction in a- and b-wave at P180. 80% and 75% reduction in a- and b-wave at P300, respectively.		Scotopic ERG absent
**Photopic ERG**	33% and 46% reduction in b-wave at P30 and P180, respectively.	79% reduction in b-wave at P30 ERG UV b-wave.	ERG b-wave comparable to WT at P30 and P180	Normal at P30 but 22% reduction in b-wave at P180.	29% reduction in b-wave at P30.	64% reduction in b-wave at P30.	60% and 80% reduction in b-wave at P180 and P300, respectively.		Photopic ERG absent
**Complex formation rods**	Mutant Prph2 unable to interact with Rom1 in rods. Intermediate and higher order complex formation slightly impaired. Rom1 occasionally found in higher order complexes.	Mutant Prph2 unable to interact with Rom1 in rods. Protein levels of Rom1 decreased No intermediate or higher order complexes were formed Prph2 and Rom1 restricted to tetramers	Prph2 could not be glycosylated and formed normal Prph2/Rom1 complexes.	Prph2 could not be glycosylated. Amount of higher order complexes decreased, while the amount of intermediate complexes was increased.	Reduction in Prph2 protein levels, less pronounced than in *Prph2^+/−^.* Rom1 protein levels not affected.	Strong decrease in Prph2 and Rom1 protein levels. No intermediate or higher order complexes were formed. Prph2 and Rom1 restricted to tetramers.			
**Protein localization**	Prph2 partially mislocalized to IS and ONL. Small amount of Rom1 mislocalized to IS.	Prph2 almost completely mislocalized to IS and ONL. Small amount of Rom1 mislocalized to IS. M- and S-Opsin as well as rhodopsin mislocalized to the IS and ONL.	None	None	None	None	NA		NA
**Fundus**	Flecking at P180, persisted till P365.	Flecking at P180, more pronounced than in C213Y/+, replaced by splotches at P365.	None	None	NA	NA	NA		NA
**Rod defect patient**	NA	NA	No patient model	NA	No patient model	NA	NA		NA
**Cone defect patient**	Pattern dystrophy	NA	No patient model	NA	No patient model	NA	NA		NA
**Reference**	[76]	[76]	[75]	[75]	[78]	[78]	[107]		[107]

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
