# Peer review of "The Interplay between Peripherin 2 Complex Formation and Degenerative Retinal Diseases"

_cells, 2020, doi:10.3390/cells9030784_

Round 1

Reviewer 1 Report

This is an interesting review of data pertaining to the photoreceptor protein prph2, found at the rims of rod and cone outer segment disks.  In general, this is a well written review suitable for publication in “Cells”.  There are some minor problems that could likely be fixed by the authors quite easily. In particular, many statements re: dominant negative should be qualified, and more explanation and care should be taken when discussing "higher order structures".

Mutations in the PRPH2 gene cause autosomal dominant RP, and may also have other phenotypes; this variability is of significant interest.  Specifically, some mutations appear to differentially affect rods or cones, despite the fact that the same gene is expressed in both cell types.  The authors review data on this protein, primarily focusing on data generated in their own studies using genetically modified mice and insights into the origins of variable phenotypes, and discussions of possible limitations of the system. The paper also contains some new data, and therefore is not entirely a review.  Specifically, Figure 1 contains H&E and EM data on C213Y, Y141C, and K153delta mice.  The mice were published in 2020, 2014, and 2016 respectively, but this figure of new data directly compares results between genotypes.  Figure 2 is similar, but presents ERG data, and Figure 3 shows fluorescence data.  In fact, the corresponding papers contain similar data - the findings are not new, but the single-figure comparison is novel; my impression is that the data has not been re-used (these are different pictures, traces, etc) but if it has been re-used, this should be stated and necessary permissions obtained.

Line 45: The difference between primary, secondary and tertiary (?) interactions is not well described.  Primary interactions are stated to be “direct”.  Are the secondary interactions indirect?  If yes, this is not stated.  If no, how are they different from primary interactions?  Is the third level actually named “higher order” (why not tertiary)?  “Higher order” is used throughout the manuscript, is never very well defined, but becomes increasingly important, which is quite frustrating to a reader.  At lines 177, a paragraph is introduced that repeats the explanation, but specific to prph2/rom1.  However, the term “intermediate” is used instead of “secondary”.  Several times, contradictory statements appear to occur. For example, on line 218 “This is a remarkable finding considering that Rom1 is normally excluded from higher order complexes” whereas on line 202: “the ability of Prph2 and Rom1 to form higher order complexes is not only relevant for maintaining the OS structure but also for its function”.  On line 605: “Rom1 is found in intermediate complexes but excluded from higher order complexes, thus it seems plausible that the higher order complexes…mediate membrane curvature”.  On line 60: “Together Prph2 and Rom1 form non-covalent tetramers and higher-order covalently linked complexes consisting of these tetramers”. These sentences seem to contradict each other as to whether rom-1 is present in “higher order complexes” or indicate that the term “higher order complexes” is not being used consistently. It is not clear how rom-1, which participates in primary and secondary interactions (? Unclear) can avoid participating in higher order complexes. Does it terminate a polymer, or similar (i.e. is rom-1 itself a dominant negative regulator of prph2)?  Overall, this seems to be a very minimal description/discussion of something very important for the understanding and readability of the review, especially as dominant negative mechanisms are frequently invoked.

Line 175:  it seems that the most likely candidate would be rom-1.  Is the phenotype of prph2-/-, rom1-/- cones known?  Virtually the same sentence appears at line 557, again is rom1 a possibility.

Line 307 and throughout: “Dominant negative” refers to a mutation that interferes with the function of the wildtype gene.  The term is used somewhat indiscriminately in the literature, when in fact, it is often very difficult to distinguish between “dominant negative” and “dominant gain of function”.  A dominant mutation is not dominant negative simply because it resembles a loss of function mutation.  Other explanations are: it IS a loss of function mutation (haploinsufficiency), or it is a dominant gain of function that has a similar phenotypic outcome.  If the end phenotype is far removed from function (e.g. “the cells are dead” or “the photoreceptors don’t work”) severity of phenotype is not a reliable measure. The possibility that prph2 mutants interfere with normal prph2 function is plausible due to the fact that the protein forms multimers, but in many cases, I believe “dominant negative” should be qualified by “consistent with dominant negative”.  One formal test not mentioned at line 307 is that overexpression of the wildtype protein should compensate for the presence of a dominant negative mutation (or haploinsufficiency), but not for a dominant gain-of function. In fact, this data is known for some prph2 mutants, but this fact is not mentioned until line 552 under discussions of gene therapy.  As noted by the authors, this is important, as gene therapy for a dominant negative could be as simple as overexpression of the WT protein, but this is unlikely to work if gain of function occurs. Therefore this should probably be mentioned at both points in the text, and it should be made clear which potentially dominant negative mutations have passed this test as each is introduced, as well as which observations are inconsistent with dominant negative.  Other mentions of “dominant negative” occur at lines 289, 337, 360, 374, 457 and 552 (possibly elsewhere).  At line 374, the data provide an alternate explanation for the dominant effect (mislocalization of prph2) that is not adequately discussed.  Similarly, at line 457, gain of function that results in abnormal higher order complexes (what are these?) would be an explanation given the description.  “Dominant negative” may be an oversimplification, as prph2 seems to participate in multiple interactions; partial dominant negatives that inhibit one prph2 activity but not others might occur, as well as mutants that are synthesized inefficiently (loss of function) but also have dominant negative activity, and mutants that do not interact with a negative regulator (rom1).

Line 341: A single codon deletion could have significantly different consequences in the context of a mouse gene vs. a human gene.  1) Presumably the deletion is in the last exon, or nonsense mediated decay would likely result in a phenotype comparable to prph2-/-.  2) If this is the case, how do the translated sequences of the mutant mouse and mutant human genes differ?  For example, if one stops immediately, while the other adds on an additional 50 amino acids, this could explain the differences observed.

Line 354: it is not immediately clear from the images why prph2C213Y/+ is stated to look “better” than prph2+/-.  Could the relevant differences be described in more detail (a few words or a sentence)?  Note that this argues against C213Y acting as dominant negative mutation – if DN, the phenotype should be more severe than prph2+/-.

Line 454: not clear…what is the nature of the abnormality…are the higher order complexes missing, or do they have an abnormal structure, if so what is known about it.

Line 457: to be consistent with dominant negative the phenotype of prph2K153delta/+ mice should be more severe than prph2+/-, but the phenotype is described as “minor structural defects”

Line 482: “exhibit to a highly variable…” should be “exhibit a highly variable…”

Line 483: “macular” should be “macula”

Table1 and 2: These tables are potentially a very useful resource.  Could the authors please add the phenotype of the prph2-/- and prph2-/+ mice to these for easier reference?

Line 548: “most patients suffer from a recessive mutation in Prph2”.  This must be a typo, earlier it is stated that Prph2 mutations cause dominant disease.

Line 597: Not clear what is meant by “complex ratio”.

Author Response

Response to Reviewer1 comments:

1) Line 45: The difference between primary, secondary and tertiary (?) interactions is not well described. Primary interactions are stated to be “direct”. Are the secondary interactions indirect? If yes, this is not stated. If no, how are they different from primary interactions? Is the third level actually named “higher order” (why not tertiary)? “Higher order” is used throughout the manuscript, is never very well defined, but becomes increasingly important, which is quite frustrating to a reader.

Response: To reduce the confusion, we have modified the text to state that the wild-type Prph2 and Rom1 are present in monomeric and dimeric forms that further assemble into homo and hetero tetrameric, octomeric and larger complexes. We refer to the primary interactions as direct while the secondary interactions as indirect. The term “higher order complexes” is restricted to wild-type Prph2 while the term “higher order aggregates” to mutant Prph2.

2) Several times, contradictory statements appear to occur. For example, on line 218 “This is a remarkable finding considering that Rom1 is normally excluded from higher order complexes” whereas on line 202: “the ability of Prph2 and Rom1 to form higher order complexes is not only relevant for maintaining the OS structure but also for its function”. On line 605: “Rom1 is found in intermediate complexes but excluded from higher order complexes, thus it seems plausible that the higher order complexes…mediate membrane curvature”. On line 60: “Together Prph2 and Rom1 form non-covalent tetramers and higher-order covalently linked complexes consisting of these tetramers”. These sentences seem to contradict each other as to whether rom-1 is present in “higher order complexes” or indicate that the term “higher order complexes” is not being used consistently.

Response: The contradictory statement in line 60 and in the other lines mentioned above by this reviewer were either deleted or modified to state that Rom1 is present in non-covalently linked hetero-tetramers and in covalently linked intermediate complexes. However, Rom1 is excluded from higher order Prph2 oligomeric complexes.

3) It is not clear how rom-1, which participates in primary and secondary interactions (? Unclear) can avoid participating in higher order complexes. Does it terminate a polymer, or similar (i.e. is rom-1 itself a dominant negative regulator of prph2)? Overall, this seems to be a very minimal description/discussion of something very important for the understanding and readability of the review, especially as dominant negative mechanisms are frequently invoked.

Response: The precise mechanisms that govern Prph2 and Rom1 complex assembly remain to be fully explored. Biochemical mechanisms for how Rom1 is sorted into only intermediate-size (and smaller) complexes are lacking. It is clear, however, from many years of data from our lab and others that Rom1 is not found in the largest Prph2 complexes “i.e. oligomers” [1-3]. Some insight may be gained from our recent work showing that Prph2 utilizes both conventional and unconventional (Golgi bypass) secretory pathways during trafficking and that when Rom1 is present, more Prph2 is trafficked conventionally [3-5]. Thus it is possible that Prph2 protein sorting during the initial steps of trafficking (e.g. in the ER and cis-Golgi) is linked to Prph2 complex assembly and the presence/absence of Rom1, but no experiments evaluating this have been done. With regard to whether Rom1 is a dominant-negative-regulator of Prph2, there is some evidence for this idea though it is not usually described in those terms. There is plentiful evidence that the ratio of Prph2 to Rom1 is essential. We have shown that excess Rom1 (in a transgenic over-expresser) is toxic to cones, but biochemical characterizations have not been done [6]. Similarly, many of the structural and functional photoreceptor defects associated with overexpression of a chimeric protein comprising the body of Rom1 and the C-terminus of Prph2 can be alleviated by knocking out Rom1 [5]. The role of Rom1 in regulating both photoreceptor structure and Prph2 function is fascinating and of ongoing interest to photoreceptor biologists.

We have modified the text to address these possibilities.

4) Line 175: it seems that the most likely candidate would be rom-1. Is the phenotype of prph2-/-, rom1-/- cones known? Virtually the same sentence appears at line 557, again is rom1 a possibility.

Response: Cone phenotype in the Prph2-/-/Rom1-/- retina has not yet been studied. Due to the minor effect of Rom1 on initialization of the cone outer segments in the Rom-/- retina, it seems unlikely that Rom1 is necessary to mediate this process.

5) Line 307 and throughout: “Dominant negative” refers to a mutation that interferes with the function of the wildtype gene. The term is used somewhat indiscriminately in the literature, when in fact, it is often very difficult to distinguish between “dominant negative” and “dominant gain of function”. A dominant mutation is not dominant negative simply because it resembles a loss of function mutation. Other explanations are: it IS a loss of function mutation (haploinsufficiency), or it is a dominant gain of function that has a similar phenotypic outcome. If the end phenotype is far removed from function (e.g. “the cells are dead” or “the photoreceptors don’t work”) severity of phenotype is not a reliable measure. The possibility that prph2 mutants interfere with normal prph2 function is plausible due to the fact that the protein forms multimers, but in many cases, I believe “dominant negative” should be qualified by “consistent with dominant negative”. One formal test not mentioned at line 307 is that overexpression of the wildtype protein should compensate for the presence of a dominant negative mutation (or haploinsufficiency), but not for a dominant gain-of function. In fact, this data is known for some prph2 mutants, but this fact is not mentioned until line 552 under discussions of gene therapy. As noted by the authors, this is important, as gene therapy for a dominant negative could be as simple as overexpression of the WT protein, but this is unlikely to work if gain of function occurs. Therefore this should probably be mentioned at both points in the text, and it should be made clear which potentially dominant negative mutations have passed this test as each is introduced, as well as which observations are inconsistent with dominant negative. Other mentions of “dominant negative” occur at lines 289, 337, 360, 374, 457 and 552 (possibly elsewhere). At line 374, the data provide an alternate explanation for the dominant effect (mislocalization of prph2) that is not adequately discussed. Similarly, at line 457, gain of function that results in abnormal higher order complexes (what are these?) would be an explanation given the description. “Dominant negative” may be an oversimplification, as prph2 seems to participate in multiple interactions; partial dominant negatives that inhibit one prph2 activity but not others might occur, as well as mutants that are synthesized inefficiently (loss of function) but also have dominant negative activity, and mutants that do not interact with a negative regulator (rom1).

Response: The term dominant negative is removed from the entire article to avoid confusion and replaced with dominant gain-of-function mutation where appropriate. Furthermore, the outcome of gene supplementation experiments was added to the mouse models for which it was performed.

Please note that at line 374, the text described the mislocalization is caused by the altered complex formation of the mutant Prph2.

6) Line 341: A single codon deletion could have significantly different consequences in the context of a mouse gene vs. a human gene. 1) Presumably the deletion is in the last exon, or nonsense mediated decay would likely result in a phenotype comparable to prph2-/-. 2) If this is the case, how do the translated sequences of the mutant mouse and mutant human genes differ? For example, if one stops immediately, while the other adds on an additional 50 amino acids, this could explain the differences observed.

Response: We agree with this reviewer for the fact that a single base pair deletion in a codon could lead to a different outcome in a mouse gene versus a human gene. The frameshift in the human gene is predicted to result in the translation of 16 additional amino acids followed by a termination codon and the formation of truncated protein that is 26 amino acid shorter while than the wildtype protein. However, such deletion in the mouse gene results in altering the last 40 amino acids of the mouse Prph2 and the addition of extra 11 amino acids. This results in the translation of 51 amino acids after codon 307. Since retinal phenotype in patients is significantly milder than the mouse phenotype, we predict that this difference is likely a consequence of the different toxic effect of these two mutant proteins on the retina. We expanded the text to address the differences between human and mouse mutations and provided potential explanation to the different phenotypic outcomes.

7) Line 354: it is not immediately clear from the images why prph2C213Y/+ is stated to look “better” than prph2+/-. Could the relevant differences be described in more detail (a few words or a sentence)? Note that this argues against C213Y acting as dominant negative mutation – if DN, the phenotype should be more severe than prph2+/-.

Response: A short description of the relevant differences was added to the text.

It is true that the structural improvement argues against dominant effect of the C213Y mutation on the structural level. On the other hand this structural improvement is opposed by a functional decline of the Prph2C213Y/+ when compared with Prph2+/- as seen in both scotopic and photopic ERGs. The functional decline argues for a dominant effect on the functional level.

8) Line 454: not clear…what is the nature of the abnormality…are the higher order complexes missing, or do they have an abnormal structure, if so what is known about it.

Response: To avoid confusion, we removed this sentence and expanded on the text to address the work presented in the publication associated with this model.

9) Line 457: to be consistent with dominant negative the phenotype of prph2K153delta/+ mice should be more severe than prph2+/-, but the phenotype is described as “minor structural defects”.

Response: Like for the C213Y mutation, the structural improvement is opposed by a functional decline when comparing Prph2K153Δ/+ and Prph2+/-. Scotopic ERGs were significantly reduced at P30, which argues for a dominant effect on the functional level.

10) Line 482: “exhibit to a highly variable…” should be “exhibit a highly variable…”

Response: Typo was corrected.

11) Line 483: “macular” should be “macula”.

Response: Typo was corrected.

12) Table1 and 2: These tables are potentially a very useful resource. Could the authors please add the phenotype of the prph2-/- and prph2-/+ mice to these for easier reference?

Response: Prph2+/- and Prph2-/- were added to the tables.

13) Line 548: “most patients suffer from a recessive mutation in Prph2”. This must be a typo, earlier it is stated that Prph2 mutations cause dominant disease.

Response: Typo was corrected.

14) Line 597: Not clear what is meant by “complex ratio”.

Response: We modified the text to include further explanation for the importance of the different complexes of Prph2.

References:

  1. Goldberg, A.F.; Molday, R.S. Subunit composition of the peripherin/rds-rom-1 disk rim complex from rod photoreceptors: hydrodynamic evidence for a tetrameric quaternary structure. Biochemistry 1996, 35, 6144-6149, doi:10.1021/bi960259n.
  2. Kevany, B.M.; Tsybovsky, Y.; Campuzano, I.D.; Schnier, P.D.; Engel, A.; Palczewski, K. Structural and functional analysis of the native peripherin-ROM1 complex isolated from photoreceptor cells. The Journal of biological chemistry 2013, 288, 36272-36284, doi:10.1074/jbc.M113.520700.
  3. Stuck, M.W.; Conley, S.M.; Naash, M.I. PRPH2/RDS and ROM-1: Historical context, current views and future considerations. Prog Retin Eye Res 2016, 52, 47-63, doi:10.1093/hmg/ddy359.
  4. Tian, G.; Ropelewski, P.; Nemet, I.; Lee, R.; Lodowski, K.H.; Imanishi, Y. An unconventional secretory pathway mediates the cilia targeting of peripherin/rds. The Journal of neuroscience : the official journal of the Society for Neuroscience 2014, 34, 992-1006, doi:10.1523/jneurosci.3437-13.2014.
  5. Conley, S.M.; Stuck, M.W.; Watson, J.N.; Zulliger, R.; Burnett, J.L.; Naash, M.I. Prph2 initiates outer segment morphogenesis but maturation requires Prph2/Rom1 oligomerization. Hum Mol Genet 2019, 28, 459-475, doi:10.1093/hmg/ddw408.
  6. Chakraborty, D.; Conley, S.M.; Nash, Z.; Ding, X.Q.; Naash, M.I. Overexpression of ROM-1 in the cone-dominant retina. Advances in experimental medicine and biology 2012, 723, 633-639, doi:10.1007/978-1-4614-0631-0_80.

Reviewer 2 Report

The review by Tebbe and colleagues summarize 30 years of research on peripherin, a tetraspanin protein found specifically in photoreceptors and involved in formation and structural integrity of the outer segment compartment. Human mutations in peripherin are associated with autosomal dominant retinitis pigmentosa and macular degeneration. In addition to covering outer segment formation, the authors spend much of the review covering animal models of different human disease mutations.  Overall the topic of peripherin’s role in outer segment formation is very timely, as recent new findings on the underlying molecular mechanism have spurred strong interest in this field. While the authors of this review include these recent findings, they fail to apply this new knowledge to previous work and therefore continue to propagate the misnomer that peripherin’s molecular function to retain membrane material for outer segment formation is not happening in cones.  

A recent finding [31] published in 2017, showed that peripherin plays a critical role in retaining membrane material, in the form of ectosomes, to the photoreceptor cilia. The previous papers studying cone outer segment formation in the peripherin knockout mouse model (rds and Nrl/rds models), [68-69], were published before the more recent finding regarding ectosome retention. These older papers do show S-opsin expression in membrane material present in the subretinal space; however, no experiment was done to equivocally show that this membrane material is connected (ie retained) to cilium in cones. Now that peripherin’s role in ectosome retention is known [31] these experiments should be done; however, until then suggesting that peripherin does not perform ectosome retention in cones is highly mis-leading. Whether peripherin’s structural role/expression level/oligomer formation within the open disc structure of cones is different from rods is compelling, and the authors are welcome to highlight those interesting observations. However, the blanket statement that COS are not formed through an ectosome-retention mechanism has yet to be tested and should be removed from the manuscript:

Line 19-20. “The importance of Prph2 for photoreceptor development, maintenance and function is underscored by the fact that its absence results in a failure to initialize OS formation in rods and formation of severely disorganized OS membranous structures in cones.”

Line 168-169. “Here the cones in the double knockout mouse (Prph2/Nrl) displayed disorganized COSs lacking flattened lamellae characteristic of WT COS.”

Line 171-172. “The ability of Prph2 retinas to form disorganized COSs argues for a different role of Prph2 in cones than in rods”

Line 175-176. “A question which remains to be answered is which protein in the developing COS is required to initialize the expansion of COS membranes?”

Line 234. “and severely disorganized COS,”

Line 249-250. “These results further highlight the differential functional roles of Prph2 in rods versus cones.”

Line 573-574. “, there seems to be distinct differences in its function in these two types of photoreceptor cells.”

Line 578-585. “The peripherin mediated suppression of ectosomes shedding from photoreceptor cilium was found to be a prerequisite for the formation of ROS. Thus far there is no study demonstrating a similar mechanism in the formation of COS. The formation of ectosomes and membrane dynamics in the formation of the COS in WT and Prph2 mice thus might be a potential approach for further studies aiming to unravel the different function of Prph2 in ROS vs COS. While the function of Prph2 in the initialization of ROS and COS formation varies, it seems to be indispensable for the correct shaping, sizing and stacking of discs and lamellae in both.”

In summary, this manuscript is a thorough review of the small tetraspanin protein, peripherin, and the current mouse models available to understand human mutations associated with peripherin. However, the authors do not integrate the current understanding of peripherin’s molecular functions with past publications. Until that is resolved I cannot recommend this review for publication.

Author Response

Reviewer 2:

1) The review by Tebbe and colleagues summarize 30 years of research on peripherin, a tetraspanin protein found specifically in photoreceptors and involved in formation and structural integrity of the outer segment compartment. Human mutations in peripherin are associated with autosomal dominant retinitis pigmentosa and macular degeneration. In addition to covering outer segment formation, the authors spend much of the review covering animal models of different human disease mutations. Overall the topic of peripherin’s role in outer segment formation is very timely, as recent new findings on the underlying molecular mechanism have spurred strong interest in this field. While the authors of this review include these recent findings, they fail to apply this new knowledge to previous work and therefore continue to propagate the misnomer that peripherin’s molecular function to retain membrane material for outer segment formation is not happening in cones.

Response: A concluding statement was added which states that the mechanism of the initialization of cone outer segment is not yet fully understood. The statement mentions especially the presence of ectosomes, to which degree membrane retention is happening and if it is mediated by Prph2 as questions which need to be addressed in further studies.

Reviewer 3 Report

In the article “The interplay between peripherin 2 complex formation and degenerative retinal diseases”, the authors, Lars Tebbe et. al. aim to provide an overall view of Prph2 function in the development of rod and cone outer segments and present a review about the different animal models expressing pathogenic Prph2 mutation. Below, please you will find several issues that I think need to be addressed:

  1.   The methods section is organized like an experimental article, this format does not apply for a Review article. It must be modified following the criteria of a review.

  1. I would consider it more appropriate to indicate the source from which the figures have been extracted in the figure caption.
  2. Figures 1, 2 and 3 provide very interesting information. But, why is there no image of all models discussed in the article?
  3. Tables 1 and 2 summarize the phenotypes. In these tables, additional models that are not commented in the review appear. What is the reason that they are only added to the table? This could generate confusion.
  4. In line 619 there is a contraction form, in my opinion, it is not correct in a formal article.

Author Response

Reviewer 3:

1) The methods section is organized like an experimental article, this format does not apply for a Review article. It must be modified following the criteria of a review.

Response: This has been modified to fit the context of a review, where it is described within the text, not its own independent section of the article.

2) I would consider it more appropriate to indicate the source from which the figures have been extracted in the figure caption.

Response: These figures were created for the review, and have not been published elsewhere, therefore we did not source the location. However we cite the methods, which did come from previous publications. We also included how the experiments were completed within the figure legend.

3) Figures 1, 2 and 3 provide very interesting information. But, why is there no image of all models discussed in the article?

Response: We included these specific genotypes in the figure because these represent the patient Prph2 mutations of the heterozygous models. Although there are other models mentioned in the article, we wanted to emphasize these select few.

4) Tables 1 and 2 summarize the phenotypes. In these tables, additional models that are not commented in the review appear. What is the reason that they are only added to the table? This could generate confusion.

Response: The genotypes that were missing have been added to the text such as Prph2N229S/+, Prph2N229S/N229s, and noted in the text where the other genotypes were presented within the paper. We also added Prph2+/- and Prph2-/- to make the table more comprehensive.

5) In line 619 there is a contraction form, in my opinion, it is not correct in a formal article.

Response: The contraction form was removed.
